# TFD: Spectrally Guided Time-Frequency Diffusion Model For Time Series Imputation

## Abstract

Diffusion models have recently shown strong advantages in generative modeling and time-series tasks thanks to its capability of accurately capturing complex data distributions through progressive denoising. However, existing time-domain only diffusion models struggle to reconstruct high-frequency details due to non-uniform energy distribution in real-world time series data. To balance the reconstruction of global trends and local dynamics, we introduce **TFD**, spectrally guided **T**ime-**F**requency **D**iffusion model for time series imputation. TFD is a hybrid time-frequency diffusion framework that couples time and frequency domain diffusion processes to achieve coarse-to-fine reconstruction, and we formalize both the noise-injection and denoising procedures following the DDPM framework. In the proposed hybrid framework, the time-domain diffusion stabilizes low-frequency trends by capturing temporal dependencies, while the frequency-domain counterpart leverages band-separable spectral representations to refine high-frequency details. We further analyze the correspondence between denoising steps and spectral components. Based on this observation, we design a frequency-aware high-pass timestep embedding that serves as spectral guidance, emphasizing the relevant bands at specific steps and enabling more accurate band-wise reconstruction. Extensive experiments demonstrate that our proposed TFD achieves state-of-the-art results across multiple benchmark datasets.

## 1 Introduction

Diffusion models (Ho et al., 2020; Song et al., 2021b; Sohl-Dickstein et al., 2015) have emerged as a powerful class of generative frameworks for time series modeling, owing to their remarkable ability to approximate complex data distributions. By progressively denoising latent representations initialized from Gaussian noise, these models learn a reverse diffusion process that effectively captures intricate temporal dependencies. Diffusion-based approaches have demonstrated state-of-the-art performance across a range of tasks, including time series imputation (Tashiro et al., 2021; Alcaraz & Strodthoff, 2023), forecasting (Rasul et al., 2021; Kollovieh et al., 2023) and generation (Yuan & Qiao, 2024; Coletta et al., 2023).

However, existing diffusion models for time series still suffer from several notable limitations: (i) **Insufficient reconstruction of high-frequency details.** (Galib et al., 2024; Yang et al., 2024; Falck et al., 2025) In the time-domain diffusion process, each step injects isotropic Gaussian noise with uniformly distributed energy. However, the energy distribution of time series is highly unbalanced and typically concentrated in the low-frequency range. High-frequency components is the first to be destroyed by noise, while low-frequency counterparts remain partially preserved in the diffusion process, making the reconstruction of high-frequency components much more difficult and thus limiting the model's ability to capture fine-grained details. (ii) **Difficulty in balancing global trends and local dynamics.** (Crabbé et al., 2024; Shen et al., 2024; Naiman et al., 2024) Global trends and local dynamics often conflict in the diffusion reconstruction process due to imbalanced energy in time series data. Emphasizing long-term trends tends to cause over-smoothing, while focusing on short-term fluctuations may disrupt global structural consistency, resulting in overall distortion. (iii) **Frequency-insensitive denoising.** (Teng et al., 2024; Yang et al., 2023) Current step embedding methods are not capable of capturing the relationship between target frequency and diffusion step during the denoising process. Since different frequency bands exhibit distinct recovery trajectories, a single-step embedding cannot provide frequency-sensitive modulation. As a result, the model treats

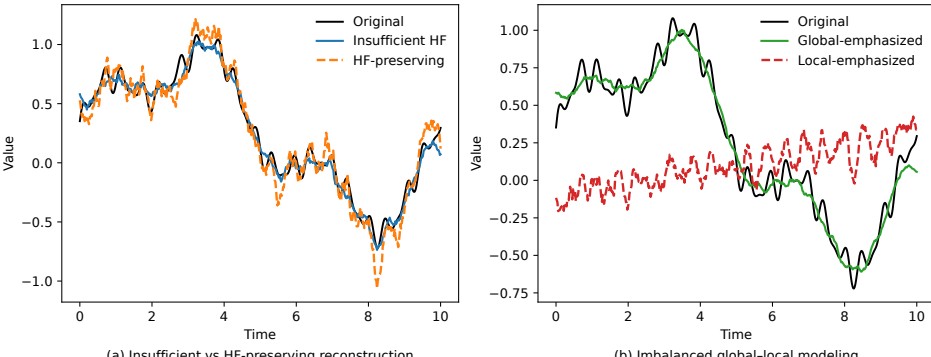

Figure 1: An illustration of Limitations (i) and (ii): In (a), the reconstructed series largely matches the ground-truth trend but is highly inaccurate in fine details compared with high-frequency preserving reconstruction methods, revealing the insufficient modeling of high-frequency information during reconstruction. In (b), emphasizing global information yields a more accurate overall pattern but poor local details, whereas emphasizing local information preserves short-term variations but causes large discrepancies in the overall structure, which highlights the imbalance between global and local information.

all frequency bands uniformly during denoising, making it difficult to allocate modeling capacity appropriately between low- and high-frequency components. This further exacerbates the loss of high-frequency details and the inadequacy of multi-scale structure modeling.

In summary, these shortcomings can be attributed to two fundamental causes: (i) **Time-domain-only noise modeling.** Both noise injection and denoising process are only in the time domain, which leads to a uniform distribution of noise across all frequency bands. This design ignores the inherently imbalanced spectral energy of real-world time series, resulting in inaccurate modeling of high-frequency component with relatively weaker energy. (ii) **Lack of frequency-specific attention in step embeddings.** Current diffusion step embedding methods do not provide frequency-specific modulation at particular diffusion steps, which fail to capture how different frequency bands should be emphasized or suppressed as the denoising process progresses. This limitation prevents the model from adaptively allocating capacity between low- and high-frequency components, which in turn exacerbates the loss of fine-grained details and weakens multi-scale representation.

To address these issue, we propose TFD, a spectrally guided time-frequency diffusion model. Compared with conventional time-domain diffusion model, diffusion in the frequency domain enables an explicit distinction across frequency bands, thereby enhancing the reconstruction of high-frequency information and addressing the limitation of insufficient high-frequency recovery in existing models. TFD leverages the strong capability of time-domain diffusion to capture low-frequency structures, while simultaneously exploiting the advantage of frequency-domain diffusion in restoring high-frequency components, resulting in a coarse-to-fine reconstruction. We formalize the hybrid noise injection and denoising processes under a DDPM framework and provide detailed descriptions of both the training and sampling procedures. Besides, we explore the relationship between diffusion steps and frequency components of the data and introduce a novel frequency-aware high-pass step embedding method as spectral guidance for the diffusion process. This embedding more effectively captures the correspondence between diffusion steps and frequency bands, enabling the model to focus on the appropriate spectral components during denoising and thereby achieving more accurate and effective reconstruction.

Our contributions are summarized as follows: (i) We propose TFD, a time-frequency diffusion model which leverages the strength of time-domain diffusion in modeling low-frequency structures and that of frequency-domain diffusion in restoring high-frequency components, achieving a coarse-to-fine reconstruction of time series. We also develop a DDPM formulation for hybrid noise injection and denoising, and provide detailed training and sampling procedures for the time-frequency diffusion model. (ii) We propose a frequency-aware high-pass timestep embedding strategy that exploits the frequency transformation patterns in the diffusion process to assign each timestep a corresponding minimum perceivable frequency, thereby providing spectral guidance during denoising, enhancing the model's ability to reconstruct specific frequency components, and improving its capacity to

restore fine-grained details. (iii) We conduct experiments on multiple datasets under various missing rates and scenarios for the time-series imputation task, achieve state-of-the-art performance, and demonstrate the effectiveness of our method.

## 2 PRELIMINARIES

### 2.1 FREQUENCY-DOMAIN TRANSFORMATIONS

We introduce three kinds of commonly-used transformations that map a discrete-time sequence from time domain into alternative spectral representations, including the Discrete Fourier Transform (DFT), the Discrete Cosine Transform (DCT) and the Haar Discrete Wavelet Transform (DWT).

Given a real-valued discrete-time series $\mathbf{x}[n], n \in \{0, 1, \ldots, N-1\}$, DFT is defined as:

$$\tilde{\mathbf{x}}[k] = \sum_{n=0}^{N-1} \mathbf{x}[n] \cdot e^{-i2\pi \frac{k}{N} n}, \quad k = 0, 1, \ldots, N-1, \tag{1}$$

while DCT is defined as:

$$\tilde{\mathbf{x}}[k] = \sum_{n=0}^{N-1} \mathbf{x}[n] \cdot \cos\left[\frac{\pi}{N}\left(n + \tfrac{1}{2}\right)k\right], \quad k = 0, 1, \ldots, N-1, \tag{2}$$

The output of DWT consists of low-frequency components ($a[k]$) and high-frequency components ($d[k]$). In our work, we utilize the Haar Wavelet transform (Sundararajan, 2015):

$$a[k] = \frac{1}{\sqrt{2}}\left(x[2k] + x[2k+1]\right)$$
$$d[k] = \frac{1}{\sqrt{2}}\left(x[2k] - x[2k+1]\right), \tag{3}$$

where $k \in \{0, 1, \cdots, \frac{N}{2} - 1\}$ and the output of Haar-DWT is a concatenation of $a[k]$ and $d[k]$, *i.e.*, $\text{DWT}(\mathbf{x}) = \{a[0], a[1], \cdots, a[\frac{N}{2}-1], d[0], d[1], \cdots, d[\frac{N}{2}-1]\}$. The properties of the frequency-domain transformations are listed in in Tab.1.

Table 1: Comparison of properties among DFT, DCT, and Haar-DWT

| Property | DFT | DCT | Haar-DWT |
|---|---|---|---|
| Basis functions | Complex exponential ($e^{-i2\pi kn/N}$) | Cosines $\cos\left(\frac{\pi}{N}(n+\frac{1}{2})k\right)$ | Haar scaling $\phi(t)$ and wavelet $\psi(t)$ |
| Periodicity | Periodic in frequency domain | Implicit periodic extension | Compact support, localized in time |
| Symmetry | Conjugate symmetry for real signals | Even symmetry extension | Symmetric box functions (piecewise constant) |
| Linearity | Linear transform | Linear transform | Linear transform |

In this paper, we explicitly employ aforementioned frequency-domain transforms to map the input sequence into alternative spectral representations, thereby achieving diffusion process in the frequency domain, which enables high-frequency refinement and more accurate detail modeling.

### 2.2 TIME SERIES DIFFUSION IN THE FREQUENCY DOMAIN

While most existing diffusion-based approaches (Wang et al., 2023; Yan et al., 2021) for time series focus exclusively on modeling in the time domain, (Crabbé et al., 2024) explores the diffusion process in the frequency domain by introducing a noise scaling matrix $\mathbf{G}$ to the frequency-domain diffusion process. Scaling matrix $\mathbf{G}$ plays two essential roles: (i) Rescaling spectral coefficients to make frequency-domain noise isotropic across all frequency components, (ii) Ensuring energy consistency between the time and frequency domains under the frequency domain transform. For an input sequence $\mathbf{x} \in \mathbb{R}^{K \times L}$, the diag noise scaling matrix $\mathbf{G} \in \mathbb{R}^{L \times L}$ is defined as :

$$\mathbf{G} = \begin{cases} 1 & n = 1 \text{ or } l \text{ is even and } n = \frac{l}{2} \\ \frac{1}{\sqrt{2}} & \text{otherwise} \end{cases} \tag{4}$$

Consistent with (Crabbé et al., 2024), we apply the noise scaling matrix $\mathbf{G}$ to the frequency domain diffusion process for energy balance.

## 2.3 PROBLEM DEFINITION AND NOTATIONS

**Definition 2.1 (Time Series with Missing Value).** A time series with missing values is defined as $\tilde{\mathbf{X}} = (\mathbf{X}, \mathbf{M}, \mathbf{T})$, where $\mathbf{X} \in \mathbb{R}^{K \times L}$ is the observation matrix with $K$ observations at a time, which are ordered along $L$ time intervals chronologically; $\mathbf{M} \in \mathbb{R}^{K \times L}$ is an indicator matrix that indicates whether the observation at $(i, j)$ in $\mathbf{X}$ is missing or not: if the observation at position $(i, j)$ is missing, $\mathbf{M}_{i,j} = 1$, otherwise, $\mathbf{M}_{i,j} = 0$; $\mathbf{T} \in \mathbb{R}^L$ is the time stamps of the time series.

**Problem Statement (Time Series Imputation).** Given a time series with missing value $\tilde{\mathbf{X}} = (\mathbf{X}, \mathbf{M}, \mathbf{T})$, the goal of time series imputation is to learn an imputation function $\mathcal{M}_\theta$, such that

$$\bar{\mathbf{X}} = \mathcal{M}_\theta(\tilde{\mathbf{X}}), \tag{5}$$

where $\bar{\mathbf{X}} \in \mathbb{R}^{K \times L}$ is the imputed time series, $\bar{\mathbf{X}}_{i,j}$ denotes the imputation output if $\mathbf{M}_{i,j} = 1$, otherwise $\bar{\mathbf{X}}_{i,j} = \tilde{\mathbf{X}}_{i,j}$.

**Notations.** We adopt the following notations. We use superscripts to denote the domain of the data, and subscripts to denote denoising steps, *e.g.,* $\mathbf{x}_k^f$ refers to the data at diffusion step $k$ in the frequency domain and $\epsilon_k^t$ refers to the noise at diffusion step $k$ in the time domain. $\beta$ is a predefined scheduler over $T$ denoising steps and $\alpha_k = 1 - \beta_k, k \in \{1, 2, \cdots, T\}$. We denote $\bar{\alpha}_{i:j} = \Pi_{k=i}^j \alpha_k$ and if $i = 1$, we omit the lower index $i$ and use $\bar{\alpha}_j$ for short. Specially, we define $\bar{\alpha}_{0:j} = 1$ for arbitrary $j$. Besides, for time series data, we following the same notations as CSDI, *i.e.,* $\mathbf{X}^{\text{ta}}$, $\mathbf{X}^{\text{obs}}$ denotes the imputation targets and ground truth values and $\mathbf{M}^{\text{cond}}$ denotes the condition matrix.

## 3 METHODOLOGY

### 3.1 TIME-FREQUENCY DIFFUSION MODEL

#### 3.1.1 FORWARD PROCESS OF TFD

The forward process of TFD consists of two successive parts, *i.e.,* noise is first injected in the frequency domain and subsequently in the time domain. Through these two successive noise injection procedures, the original data distribution $p(x_0)$ is transformed into a target Gaussian distribution. The first part introduces frequency-domain noise, for a given transformation $\mathcal{F}$, we first map the data into corresponding domain via $\mathcal{F}$, and then inject random gaussian noise $\epsilon_k^f \sim \mathcal{N}(0, \mathbf{I})$ into this frequency domain representation in a DDPM-like style. This process is formulated as:

$$\mathbf{x}_k^f = \sqrt{\alpha_k^f} \mathcal{F}(\mathbf{x}_{k-1}^t) + \sqrt{\beta_k^f} \sqrt{1-\lambda}(\mathbf{G}\epsilon_k^f), \qquad \epsilon_k^f \sim \mathcal{N}(0, \mathbf{I}) \tag{6}$$

where $\lambda$ is the coefficient for balancing variance between frequency and time domain noise. Then we transform the frequency-domain noisy sample $\mathbf{x}_k^f$ in Eq.6 back to the time domain:

$$\mathcal{F}^{-1}(\mathbf{x}_k^f) = \sqrt{\alpha_k^f} \mathbf{x}_{k-1}^t + \sqrt{\beta_k^f} \sqrt{1-\lambda} \mathcal{F}^{-1}(\mathbf{G}\epsilon_k^f), \qquad \epsilon_k^f \sim \mathcal{N}(0, \mathbf{I}) \tag{7}$$

leading to the posterior of $q(\mathcal{F}^{-1}(\mathbf{x}_k^f)|\mathbf{x}_{k-1}^t) := \mathcal{N}(\sqrt{\alpha_k^f} \mathbf{x}_{k-1}^t, \beta_k^f(1-\lambda)\mathcal{F}^{-1}\mathbf{G}\mathbf{G}^T(\mathcal{F}^{-1})^T)$. Then **another** independent gaussian noise $\epsilon_k^t$ is injected to $\mathcal{F}^{-1}(\mathbf{x}_k^t)$, *i.e.,* noise is injected in the time domain:

$$\mathbf{x}_k^t = \sqrt{\alpha_k^t} \mathcal{F}^{-1}(\mathbf{x}_k^f) + \sqrt{\beta_k^t} \sqrt{\lambda} \epsilon_k^t, \qquad \epsilon_k^t \sim \mathcal{N}(0, \mathbf{I}) \tag{8}$$

leading to the posterior of $q(\mathbf{x}_k^t|\mathcal{F}^{-1}(\mathbf{x}_k^f)) := \mathcal{N}(\mathcal{F}^{-1}(\mathbf{x}_k^f), \beta_k \lambda \mathbf{I})$. Thus the noise injection loop $k$ is completed and we can model the relationship between $\mathbf{x}_k^t$ and $\mathbf{x}_{k-1}^t$ as:

$$\mathbf{x}_k^t = \sqrt{\alpha_k^t \alpha_k^f} \mathbf{x}_{k-1}^t + \sqrt{\alpha_k^t(1-\alpha_k^f)} \sqrt{1-\lambda} \mathcal{F}^{-1}(\mathbf{G}\epsilon_k^f) + \sqrt{1-\alpha_k^t} \sqrt{\lambda} \epsilon_k^t \tag{9}$$

By iteratively preforming the noise injection process according to Eq.7, Eq.8 and Eq.9, we can model the relationship between $\mathbf{x}_k^t$ and $\mathbf{x}_0^t$ as:

$$\mathbf{x}_k^t = \sqrt{\bar{\alpha}_k^t \bar{\alpha}_k^f} \mathbf{x}_0^t + \sqrt{1-\lambda} \sum_{s=1}^k \sqrt{\beta_s^f} \sqrt{\frac{\bar{\alpha}_k^t}{\bar{\alpha}_{s-1}^t}} \sqrt{\frac{\bar{\alpha}_k^f}{\bar{\alpha}_s^f}} \mathcal{F}^{-1}(\mathbf{G}\epsilon_s^f) + \sqrt{\lambda} \sum_{s=1}^k \sqrt{\beta_s^t} \sqrt{\frac{\bar{\alpha}_k^t}{\bar{\alpha}_s^t}} \epsilon_s^t \tag{10}$$

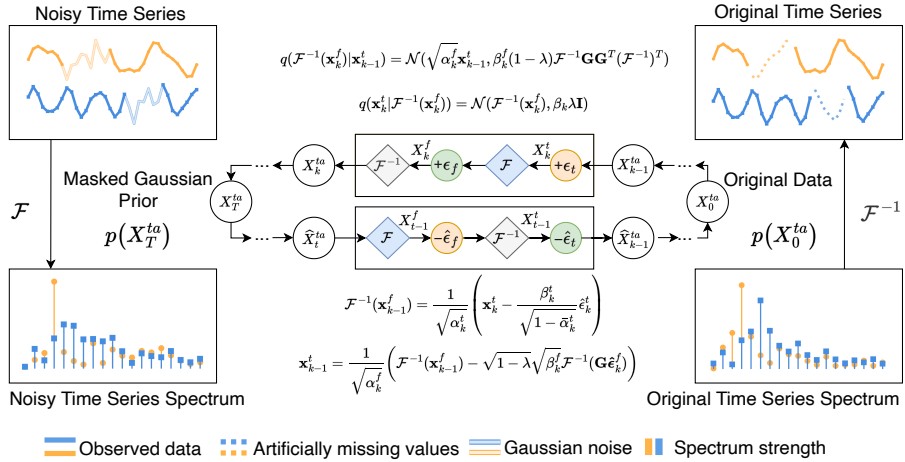

Figure 2: Noise injection and denoising process in TFD. At each forward step, frequency-domain noise is injected first and then time-domain noise. In the reverse process, time-domain noise is removed first, followed by frequency-domain noise.

According to Eq.10, for arbitrary $k$, $\mathbf{x}_k^t$ is a linear transformation of $\mathbf{x}_0^t$ with two groups of independent gaussian noise $\boldsymbol{\epsilon}_s^t$ and $\boldsymbol{\epsilon}_s^f$, $s \in \{1, 2, \cdots, k\}$. Therefore, the posterior $q(\mathbf{x}_k^t|\mathbf{x}_0^t)$ remains gaussian, *i.e.,*

$$q(\mathbf{x}_k^t|\mathbf{x}_0^t) := \mathcal{N}(\boldsymbol{\mu}_k, \boldsymbol{\Sigma}_k), \tag{11}$$

where $\boldsymbol{\mu}_k = \sqrt{\bar{\alpha}_k^t \bar{\alpha}_k^f} \mathbf{x}_0^t$, $\boldsymbol{\Sigma}_k = (1-\lambda) \sum_{s=1}^k \beta_s^f \frac{\bar{\alpha}_k^t}{\bar{\alpha}_{s-1}^t} \frac{\bar{\alpha}_k^f}{\bar{\alpha}_s^f} \left(\mathcal{F}^{-1}\mathbf{G}\mathbf{G}^T(\mathcal{F}^{-1})^T\right) + \lambda \sum_{s=1}^k \beta_s^t \frac{\bar{\alpha}_k^t}{\bar{\alpha}_s^t} \mathbf{I}$.

### 3.1.2 REVERSE PROCESS OF TFD

To progressively reconstruct the original data from the diffused samples, we construct a reverse Markov chain that strictly corresponds to the forward noise injection process. We parameterize the reverse distribution $p_\theta(\mathbf{x}_{k-1}|\mathbf{x}_k)$ via a Gaussian approximation, aiming to model the true posterior $q(\mathbf{x}_{k-1}|\mathbf{x}_k, \mathbf{x}_0)$. This enables closed-form stepwise sampling throughout the denoising trajectory. Since our method introduce a combination of time-domain noise and frequency-domain noise (Eq.9) at each diffusion step, the reverse process must accurately capture the structure of the composite noise to ensure faithful data reconstruction.

To improve sampling stability, we adopt a DDIM-style sampling (Song et al., 2021a) with a deterministic trajectory in the reverse denoising process. We initialize from a Gaussian prior that is consistent with the terminal distribution of the forward diffusion. As formalized in Eq.11, the sampling process starts from $\mathbf{x}_T = \sqrt{\lambda}\boldsymbol{\epsilon}^t + \sqrt{1-\lambda}\mathcal{F}^{-1}(\mathbf{G}\boldsymbol{\epsilon}^f)$, $\boldsymbol{\epsilon}^t, \boldsymbol{\epsilon}^f \sim \mathcal{N}(0, \mathbf{I})$. At each timestep $k$, we first obtain the time domain noise estimation $\hat{\boldsymbol{\epsilon}}_k^t$ and frequency domain noise estimation $\hat{\boldsymbol{\epsilon}}_k^f$. First, the time-domain noise in $\mathbf{x}_k^t$ is removed according to:

$$\mathcal{F}^{-1}(\mathbf{x}_{k-1}^f) = \frac{1}{\sqrt{\alpha_k^t}} \left( \mathbf{x}_k^t - \frac{\beta_k^t}{\sqrt{1-\bar{\alpha}_k^t}} \hat{\boldsymbol{\epsilon}}_k^t \right) \tag{12}$$

Then we remove the frequency-domain noise in $\mathcal{F}^{-1}(\mathbf{x}_{k-1}^f)$ to obtain the updated time-domain sample $\mathbf{x}_{k-1}^t$:

$$\mathbf{x}_{k-1}^t = \frac{1}{\sqrt{\alpha_k^f}} \left( \mathcal{F}^{-1}(\mathbf{x}_{k-1}^f) - \sqrt{1-\lambda}\sqrt{\beta_k^f}\mathcal{F}^{-1}(\mathbf{G}\hat{\boldsymbol{\epsilon}}_k^f) \right) \tag{13}$$

Eq.12 and Eq.13 presents different update rules for time and frequency denoising process, reflecting an parameterization choice. For simplicity, we re-parameterize the time-domain target as $\sqrt{\lambda}\hat{\boldsymbol{\epsilon}}_k^t$, so $\lambda$ does not appear explicitly in Eq.12. In contrast, the frequency branch predicts raw spectral noise $\hat{\boldsymbol{\epsilon}}_k^f$, therefore, the weight $\sqrt{1-\lambda}$ remains. To summarize, the cascaded denoising strategy *i.e.,* time-to-frequency denoising, serves as the mirror counterpart of the forward diffusion process

(frequency-to-time noise injection). Empirically, the time-domain branch is effective in stabilizing global structures and low-frequency trends, while the frequency-domain branch focuses on high-frequency components. Our design realizes a coarse-to-fine denoising mechanism. To decouple the two branches, we detach gradients from the time-domain output when entering the frequency-domain stage. The forward and reverse process is illustrated in Fig.2.

### 3.1.3 LOSS FUNCTION

As stated in Sec.3.1.1 and Sec.3.1.2, the overall noise injection process consists of two successive sub-processes, *i.e.,* time and frequency domain noise injection. We adopt the commonly used noise estimation approach in diffusion models. The loss function consists of three components: (i) time domain noise estimation loss, *i.e.,* the $l_2$ distance between the estimated time domain noise and forward time domain noise, (ii) frequency domain noise estimation loss, *i.e.,* the $l_2$ distance between the estimated frequency domain noise and forward frequency domain noise. (iii) consistency loss, *i.e.,* the $l_2$ distance between total noise from step $k$ to step $k-1$. This consistency term mitigates potential mismatches arising from estimating the two branches separately and discourages arbitrary residual reallocation across branches. Accordingly, the overall loss is defined as:

$$\mathcal{L}^d = \mathbb{E}_{\mathbf{x}_0 \sim q(\mathbf{x}_0), k} \left( \|\boldsymbol{\epsilon}_k^t - \hat{\boldsymbol{\epsilon}}_k^t\|_2^2 + \|\boldsymbol{\epsilon}_k^f - \mathcal{F}^{-1}(\mathbf{G}\hat{\boldsymbol{\epsilon}}_k^f)\|_2^2 + \omega\|(\boldsymbol{\epsilon}_k^t + \boldsymbol{\epsilon}_k^f) - (\hat{\boldsymbol{\epsilon}}_k^t + \mathcal{F}^{-1}(\mathbf{G}\hat{\boldsymbol{\epsilon}}_k^f))\|_2^2 \right),$$
(14)

where $w$ is the weight for consistency loss. In addition, since our model is specifically designed for time series imputation task, we focus solely on the reconstruction error over masked regions during training, *i.e.,* $\mathcal{L} = \mathbf{M}^t \odot \mathcal{L}^d$. The detailed training and sampling algorithm is presented in Alg.1 and 2.

---

**Algorithm 1** Training Procedure of TFD

1: **Input:** Observed sequence $\mathbf{x}_0$, condition mask $\mathbf{M}^{\text{cond}}$, observation mask $\mathbf{M}^{\text{obs}}$, side information $\mathbf{s}$, number of steps $T$, number of iterations $N$, time- and frequency-domain scheduler $\beta^t, \beta^f$, total denoising step $T$, transformation $\mathcal{F}$, noise balance coefficient $\lambda$, consistency weight $\omega$.
2: **Output:** Denoising function $\epsilon_\theta$
3: **for** $i = 1$ **to** T **do**
4:      $k \sim \text{Uniform}(\{1, 2, \cdots, T\})$
5:      Calculate time-domain noise: $\boldsymbol{\epsilon}_k^t = \sqrt{\lambda} \sum_{s=1}^k \sqrt{\beta_s^t} \sqrt{\frac{\bar{\alpha}_k^t}{\bar{\alpha}_s^t}} \boldsymbol{\epsilon}_s^t, \quad \boldsymbol{\epsilon}_s^t \sim \mathcal{N}(0, \mathbf{I}).$
6:      Calculate frequency-domain noise:

       $\boldsymbol{\epsilon}_k^f = \sqrt{1 - \lambda} \sum_{s=1}^k \sqrt{\beta_s^f} \sqrt{\frac{\bar{\alpha}_k^t}{\bar{\alpha}_{s-1}^t}} \sqrt{\frac{\bar{\alpha}_k^f}{\bar{\alpha}_s^f}} \mathcal{F}^{-1}(\mathbf{G}\boldsymbol{\epsilon}_s^f), \quad \boldsymbol{\epsilon}_s^f \sim \mathcal{N}(0, \mathbf{I}).$

7:      Calculate noisy sample at step $k$: $\mathbf{x}_k^t = \sqrt{\alpha_k^t \alpha_k^f} \mathbf{x}_0^t + \boldsymbol{\epsilon}_k^t + \boldsymbol{\epsilon}_k^f$
8:      Construct time domain input: $\mathbf{h}^t \leftarrow \texttt{set\_input}(\mathbf{x}_k^t, \mathbf{X}^{\text{obs}}, \mathbf{M}^{\text{cond}})$
9:      Estimate time domain noise and calculate time noise loss:
       $\hat{\boldsymbol{\epsilon}}_{k,\_}^t = \boldsymbol{\epsilon}_\theta(\mathbf{h}_t, \mathbf{s}, k), \mathcal{L}^t = \|\hat{\boldsymbol{\epsilon}}_k^t - \boldsymbol{\epsilon}_k^t\|_2^2$
10:     Update and construct frequency domain input: $\mathcal{F}^{-1}(\mathbf{x}_k^f) = \frac{1}{\sqrt{\alpha_k^t}} \left( \mathbf{x}_k^t - \frac{\beta_k^t}{\sqrt{1-\bar{\alpha}_k^t}} \hat{\boldsymbol{\epsilon}}_k^t \right)$
11:     Construct frequency domain input: $\mathbf{h}^f = \texttt{set\_input}(\mathcal{F}^{-1}(\mathbf{x}_k^f), \mathbf{s}, k)$
12:     Estimate frequency domain noise and calculate frequency noise loss:
       $\_, \hat{\boldsymbol{\epsilon}}_k^f = \hat{\boldsymbol{\epsilon}}_\theta(\mathbf{h}^f, \mathbf{s}, k), \mathcal{L}^f = \|\mathcal{F}^{-1}(\mathbf{G}\hat{\boldsymbol{\epsilon}}_k^f) - \boldsymbol{\epsilon}_k^f\|_2^2$
13:     Calculate consistency loss: $\mathcal{L}_c = \|(\boldsymbol{\epsilon}_k^t + \boldsymbol{\epsilon}_k^f) - (\hat{\boldsymbol{\epsilon}}_k^t + \mathcal{F}^{-1}(\mathbf{G}\hat{\boldsymbol{\epsilon}}_k^f))\|_2^2$
14:     Take gradient descent step on $\nabla_\theta \left[ (1 - \mathbf{M}^{\text{cond}}) \cdot (\mathcal{L}^t + \mathcal{L}^f + \omega\mathcal{L}^c) \right]$
15: **end for**
16: **return** $\epsilon_\theta$

---

### 3.2 FREQUENCY-AWARE HIGH-PASS DIFFUSION EMBEDDING

Considering the relationship between denoising steps and the frequency components of the data, we present the following proposition (Ning et al., 2024; Chen et al., 2025):

---

**Algorithm 2** Sampling Procedure of TFD

---

1: **Input:** Trained denoising function $\epsilon_\theta$, sampling step $T$, observed data $\mathbf{X}^{\text{obs}}$, condition mask $\mathbf{M}^{\text{cond}}$, side information $\mathbf{s}$, time domain scheduler $\beta^t$, frequency domain scheduler $\beta^f$
2: **Output:** Generated Sequence $\mathbf{x}_0$
3: Initialize by sampling from noise prior: $\mathbf{x}_T = \sqrt{\lambda}\mathbf{z}^t + \sqrt{1-\lambda}\mathcal{F}^{-1}(\mathbf{G}\mathbf{z}^f). \quad \mathbf{z}^t, \mathbf{z}^f \sim \mathcal{N}(0, \mathbf{I})$
4: **for** $k = T$ **to** $1$ **do**
5:     Construct time-domain input: $\mathbf{h}_k^t \leftarrow \mathtt{set\_input}(\mathbf{x}_k^t, \mathbf{X}^{\text{obs}}, \mathbf{M}^{\text{cond}})$
6:     Estimate time-domain noise: $\hat{\boldsymbol{\epsilon}}_k^t, \_ \leftarrow \boldsymbol{\epsilon}_\theta(\mathbf{h}_k^t, \mathbf{s}, k)$
7:     Update input according to time-domain noise: $\mathcal{F}^{-1}(\mathbf{x}_k^f) = \frac{1}{\sqrt{\alpha_k^t}}\left(\mathbf{x}_k - \frac{\beta_k^t}{\sqrt{1-\bar{\alpha}_k^t}}\hat{\boldsymbol{\epsilon}}_k^t\right)$
8:     Construct frequency-domain input: $\mathbf{h}_k^f \leftarrow \mathtt{set\_input}(\mathcal{F}^{-1}(\mathbf{x}_k^f), \mathbf{X}^{\text{obs}}, \mathbf{M}^{\text{cond}})$
9:     Estimate frequency-domain noise: $\_, \hat{\boldsymbol{\epsilon}}_k^f \leftarrow \boldsymbol{\epsilon}_\theta(\mathbf{h}_k^f, \mathbf{s}, k)$
10:    Update $\mathbf{x}_{k-1}^t \leftarrow \frac{1}{\sqrt{\alpha_k^f}}\left(\mathcal{F}^{-1}(\mathbf{x}_k^f) - \sqrt{1-\lambda}\mathcal{F}^{-1}(\mathbf{G}\hat{\boldsymbol{\epsilon}}_k^f)\right)$
11: **end for**
12: **return** $\mathbf{x}_0$

---

**Proposition 3.1** (Frequency Components in the Diffusion Process). *During the diffusion process, the corruption and reconstruction of different frequency components are non-uniform across diffusion steps. Suppose the overall diffusion process consists of $T$ steps. In the early stages of noise-injection process ($t \to 0$), the high-frequency components are corrupted first. In contrast, in the latter stages ($t \to T$), the low-frequency components are gradually corrupted. The reverse (denoising) process exhibits a corresponding behavior: the model first reconstructs the low-frequency components, followed by the reconstruction of high-frequency details.*

*Proof.* Considering a diffusion process

$$\mathrm{d}\mathbf{x}_t = \boldsymbol{f}(\boldsymbol{x}, \boldsymbol{t})\mathrm{d}t + g(t)\mathrm{d}\boldsymbol{w_t} \tag{15}$$

where $\boldsymbol{w_t}$ is the standard Wiener process in $\mathbb{R}^{\mathrm{d}_x}$, $\boldsymbol{f} : \mathbb{R}^{\mathrm{d}_{x_t}} \times [0, T] \to \mathbb{R}^{\mathrm{d}_{x_t}}$ is the drift and $g(t) : [0, T] \to \mathbb{R}^{N \times N}$ is the diffusion coefficient. The solution to Eq.15 is formulated as:

$$\mathbf{x}_t = \mathbf{x}_0 + \int_0^t \boldsymbol{f}(\mathbf{x}_s, s)\mathrm{d}s + \int_0^t g(s)\mathrm{d}\boldsymbol{w_s} \tag{16}$$

By applying DFT to Eq.16, we have:

$$\hat{\mathbf{x}}_t(\boldsymbol{\omega}) = \hat{\mathbf{x}}_0(\boldsymbol{\omega}) + \hat{f}(\boldsymbol{\omega}) + \hat{\epsilon}_t(\boldsymbol{\omega}), \tag{17}$$

where $\hat{\mathbf{x}}_t(\omega)$ is the DFT of the noisy item and satisfies: $\mathbb{E}[\hat{\epsilon}_t(\omega)] = 0$ and $\mathbb{E}[|\hat{\epsilon}_t(\omega)|^2] = \int_0^t |g(s)|^2\mathrm{d}s$ The signal-noise-ratio (SNR) of frequency $\omega$ is defined as:

$$\mathrm{SNR}(\omega) = \frac{|\hat{\mathbf{x}}_t(\omega)|^2}{\mathbb{E}[|\hat{\epsilon}_t(\omega)|^2]} = \frac{|\hat{\mathbf{x}}_t(\omega)|^2}{\int_0^t |g(s)|^2\mathrm{d}s} \tag{18}$$

Given an SNR threshold $\gamma$, we have:

$$\int_0^t |g(s)|^2\mathrm{d}s = \frac{|\hat{\mathbf{x}}_t(\omega)|^2}{\gamma} \propto \frac{\omega^{-\alpha}}{\gamma}, \tag{19}$$

where $\alpha$ is a positive constant and $|\hat{\mathbf{x}}_t(\omega)|^2 \propto \omega^{-\alpha}$ is due to low-pass characteristics of time series data. From Eq.19, it is clear that for higher frequencies ($\omega \uparrow$), the time (or step) to reach the SNR threshold decreases ($t \downarrow$), which in turn proves the above proposition. $\square$

From Prop.3.1 and Eq.19, we can calculate the denoising step to reach corresponding frequency. For simplicity, we set $g(s) = \sigma$ as a constant and thus we have:

$$\omega = (t\sigma^2\gamma)^{-\frac{1}{\alpha}}, \tag{20}$$

where $1 \leq \alpha \leq 2$, $\sigma, \gamma$ are constants. Eq.20 reveals a possible relationship between the frequency threshold and diffusion timestep, which is capable of attending to frequency-specific temporal patterns. Our frequency-aware high pass time step embedding modules serves as a high-pass filter with a frequency threshold. For denoising step $k$, the proposed embedding enables the denoising process focus on frequencies bands higher than the threshold $(k\sigma^2\gamma)^{-\frac{1}{\alpha}}$, thus provides a flexible mechanism to emphasize or de-emphasize different spectral bands depending on the current noise level, improving both stability and reconstruction fidelity in time-frequency diffusion tasks.

## 3.3 MODEL ARCHITECTURE

We present the details of the denoising network $\epsilon_\theta$. We adopt a cascaded dual-branch network as the core denoising architecture, where both branches share the same network structure. Considering the intra-channel and inter-channel dependencies in multivariate time series, which still remain after transforming to the frequency domain, the temporal-feature transformer architecture from CSDI (Tashiro et al., 2021) is adopted as the diffusion backbone.

Specifically, our model takes the noisy data, conditional information and diffusion step embedding as input. The input is first processed by a temporal attention module and a feature attention module for feature extraction, followed by a linear projection to estimate time-domain noise $\hat{\epsilon}_t$ To ensure that the frequency-domain branch focuses exclusively on estimating frequency-domain noise, we transform the output of the time-domain branch into the frequency domain via a given transformation $\mathcal{F}$ and use it as the input to the frequency-domain branch. The frequency-domain branch is symmetric to the time-domain branch, except that the step embedding is replaced with the frequency-aware high-pass diffusion embedding introduced in Sec. 3.2. Finally, the frequency-domain output is mapped back to the time domain via $\mathcal{F}^{-1}$. Overall, the model produces two outputs: the estimated time-domain noise $\hat{\epsilon}_t$ and frequency-domain noise $\hat{\epsilon}_t$. The details of our model architecture is in the Appendix A.

## 4 EXPERIMENTS

**Experimental Settings.** All our experiments are implemented in Python 3.12 using the Pytorch (Paszke et al., 2019) framework on a single Nvidia RTX 3090 GPU. We use the Adam optimizer (Kingma & Ba, 2015) with an initial learning rate of $5 \times 10^{-4}$ with a multi-step scheduler that decays the learning rate by a factor of 10 at 75% and 90% of the total training epochs. In the diffusion process, we employ a quadratic noise scheduler in both time and frequency domain diffusion.

**Datasets and Evaluation Metrics** We evaluate our results on two commonly-used real world datasets with missing data, Physionet2012 (Silva et al., 2012) and Air Quality (Yi et al., 2016). We report the Mean Absolute Error (MAE) and Root Mean Squared Error (RMSE) between model outputs and ground-truth data on the imputed positions. Please refer to the Appendix A for more details about our evaluation metrics, hyperparameters and dataset details.

**Baselines** We compare our method against a diverse set of imputation baselines, including classical statistical approaches (Mean and Lerp), deep learning methods (BRITS (Cao et al., 2018), SAITS (Du et al., 2023), TimesNet (Wu et al., 2023)), deep generative models (GP-VAE (Fortuin et al., 2020), SSGAN (Miao et al., 2021), CSDI (Tashiro et al., 2021), CSBI (Chen et al., 2023), LSCD (Fons et al., 2025)), and a time-series foundation model (ModernTCN (Luo & Wang, 2024)).

### 4.1 TIME SERIES IMPUTATION RESULTS

Table.2 shows that TFD-based methods consistently achieve state-of-the-art results across different missing ratios and datasets. On Physionet with 10% missing rate, TFD (DCT) yields the best MAE and RMSE, while TFD (DWT) and TFD (DFT) follow closely, indicating strong performance under low missingness. At 50%, TFD (DCT) attains the lowest MAE and TFD (DFT) secures the best RMSE, with TFD (DWT) ranking second on both, demonstrating robustness at moderate ratios. At 90%, TFD (DCT) again delivers the best overall results, and TFD (DFT) ranks second, surpassing all non-TFD baselines by a clear margin. On Air Quality, TFD (DFT) ranks second and TFD (DWT) remains competitive, confirming the effectiveness of TFD under non-random missingness. The corresponding visualization results are provided in the Appendix A.

Table 2: Imputation performance on Physionet and Air Quality datasets. Best results are in **bold**; second-best are underlined. ↓ indicates lower is better.

| Method | Physionet 10% | | Physionet 50% | | Physionet 90% | | Air Quality | |
|---|---|---|---|---|---|---|---|---|
| | MAE (↓) | RMSE (↓) | MAE (↓) | RMSE (↓) | MAE (↓) | RMSE(↓) | MAE (↓) | RMSE (↓) |
| Mean | 0.714 | 1.035 | 0.711 | 1.091 | 0.710 | 1.097 | 50.685 | 66.558 |
| Lerp | 0.372 | 0.708 | 0.417 | 0.840 | 0.565 | 0.993 | 15.363 | 27.658 |
| BRITS | 0.278 | 0.693 | 0.385 | 0.833 | 0.560 | 0.975 | 16.519 | 26.775 |
| GPVAE | 0.469 | 0.783 | 0.521 | 0.907 | 0.642 | 1.038 | 23.941 | 40.586 |
| SSGAN | 0.323 | 0.662 | 0.449 | 0.852 | 0.670 | 1.060 | 32.999 | 48.951 |
| TimesNet | 0.375 | 0.690 | 0.453 | 0.840 | 0.642 | 1.031 | 22.685 | 39.336 |
| CSDI | 0.219 | 0.545 | 0.307 | 0.672 | 0.481 | 0.834 | 9.670 | 19.093 |
| CSBI | 0.230 | 0.547 | 0.310 | 0.649 | – | 0.834 | 9.800 | 19.000 |
| SAITS | 0.232 | 0.583 | 0.315 | 0.735 | 0.565 | 0.971 | 15.424 | 30.558 |
| ModernTCN | 0.351 | 0.697 | 0.440 | 0.803 | 0.647 | 1.026 | 24.089 | 40.052 |
| LSCD | 0.211 | 0.494 | 0.303 | 0.664 | 0.479 | 0.832 | **9.069** | **17.914** |
| TFD (DFT) | 0.211 | 0.451 | 0.297 | **0.644** | 0.459 | 0.812 | 9.499 | 18.156 |
| TFD (DCT) | **0.208** | **0.446** | **0.296** | 0.655 | **0.456** | **0.808** | 10.586 | 21.026 |
| TFD (DWT) | 0.209 | 0.451 | 0.297 | 0.649 | 0.460 | 0.815 | 10.069 | 18.835 |

## 4.2 ABLATION STUDIES AND PARAMETER ANALYSIS

To validate the effectiveness of Frequency Aware High Pass Embedding module, we remove that with the diffusion-step module in the time diffusion process and we also evaluate the impact of key hyperparameters. The coefficient $\lambda$ controls noise allocation between the time- and frequency-domain processes; we sweep $\lambda \in \{0, 0.25, 0.5, 0.75, 1\}$ to determine the optimal allocation. We also compare different transforms $\mathcal{F} \in \{\text{DFT}, \text{DCT}, \text{DWT}\}$.

Table 3: Results of ablation studies and parameter analysis. "HP" stands for the proposed frequency-aware high pass embedding and "DE" stands for the diffusion embedding in (Tashiro et al., 2021).

| $\lambda$ | $\mathcal{F}$ | Embedding | Physionet 10% | | Physionet 50% | | Physionet 90% | |
|---|---|---|---|---|---|---|---|---|
| | | | MAE | RMSE | MAE | RMSE | MAE | RMSE |
| 0.75 | dft | HP | 0.212 | 0.495 | 0.296 | 0.664 | 0.459 | 0.812 |
| 0.75 | dft | DE | 0.214 | 0.537 | 0.311 | 0.683 | 0.478 | 0.834 |
| 0.75 | dwt | HP | 0.209 | 0.451 | 0.297 | 0.649 | 0.46 | 0.815 |
| 0.75 | dct | HP | 0.208 | 0.446 | 0.296 | 0.655 | 0.456 | 0.808 |
| 0 | dft | HP | 0.224 | 0.636 | 0.299 | 0.665 | 0.47 | 0.837 |
| 0.25 | dft | HP | 0.211 | 0.451 | 0.297 | 0.644 | 0.466 | 0.827 |
| 0.5 | dft | HP | 0.211 | 0.455 | 0.297 | 0.659 | 0.464 | 0.821 |
| 1 | dft | HP | 0.244 | 0.508 | 0.296 | 0.668 | 0.522 | 0.856 |

From Table.3, we can first observe that replacing the diffusion embedding from (Tashiro et al., 2021) with our proposed frequency-aware high-pass embedding leads to a significant improvement in both MAE and RMSE, demonstrating the effectiveness of the proposed module. Notably, compared to models that rely solely on either frequency-domain or time-domain noise ($\lambda = 0, 1$), our time-frequency hybrid model ($\lambda \neq 0, 1$) consistently achieves superior performance across all noise ratios, demonstrating the effectiveness of the proposed hybrid framework. Moreover, under different missing rates, there is no universally optimal choice of $\lambda$ or transformation $\mathcal{F}$. Therefore, the selection of model parameters should be adapted to the characteristics of the specific time series data and the corresponding missing rate.

## 5 CONCLUSION

In this paper, we propose TFD, a spectrally guided time-frequency diffusion model. By integrating time- and frequency-domain processes within a unified diffusion framework, our model enables a coarse-to-fine reconstruction mechanism. Moreover, we introduce a frequency-aware high-pass filtering diffusion-step embedding, which allows the model to focus on specific frequency components at designated steps, thereby enhancing its reconstruction capability in high frequency components. Experimental results demonstrate that our approach achieves consistently strong performance across multiple datasets and under varying missing rates.

## 6 ETHICS STATEMENT

This work adheres to the ICLR Code of Ethics. In this study, no human subjects or animal experimentation was involved. All datasets used, including Physionet2012 and Air quality, were sourced in compliance with relevant usage guidelines, ensuring no violation of privacy. We have taken care to avoid any biases or discriminatory outcomes in our research process. No personally identifiable information was used, and no experiments were conducted that could raise privacy or security concerns. We are committed to maintaining transparency and integrity throughout the research process.

## 7 REPRODUCIBILITY STATEMENT

We have made every effort to ensure that the results presented in this paper are reproducible. All code and datasets have been made publicly at `https://anonymous.4open.science/r/TFD-FCC5/`. The experimental setup, including training steps, model configurations, and hardware details, is described in detail in the paper.

Additionally, datasets used in the paper, such as Physionet2012 and Air Quality, are publicly available, ensuring consistent and reproducible evaluation results.

We believe these measures will enable other researchers to reproduce our work and further advance the field.

## 8 LLM USAGE

Large Language Models (LLMs) were used to aid in the writing and polishing of the manuscript. Specifically, we used an LLM to assist in refining the language, improving readability, and ensuring clarity in various sections of the paper. The model helped with tasks such as sentence rephrasing, grammar checking, and enhancing the overall flow of the text.

It is important to note that the LLM was not involved in the ideation, research methodology, or experimental design. All research concepts, ideas, and analyses were developed and conducted by the authors. The contributions of the LLM were solely focused on improving the linguistic quality of the paper, with no involvement in the scientific content or data analysis.

The authors take full responsibility for the content of the manuscript, including any text generated or polished by the LLM. We have ensured that the LLM-generated text adheres to ethical guidelines and does not contribute to plagiarism or scientific misconduct.

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

## A  APPENDIX

### A.1  DETAILS IN FORWARD PROCESS

**Posterior of Eq.7**: Eq.7 presents $\mathcal{F}^{-1}(\mathbf{x}_k^f) = \sqrt{\alpha_k^f}\mathbf{x}_{k-1}^t + \sqrt{\beta_k^f}\sqrt{1-\lambda}\mathcal{F}^{-1}(\mathbf{G}\boldsymbol{\epsilon}_k^f)$,   $\boldsymbol{\epsilon}_k^f \sim \mathcal{N}(0, \mathbf{I})$. $\mathcal{F}^{-1}(\mathbf{x}_{k-1}^t)$ is a linear transformation of $\mathbf{x}_k^t$ along with a Gaussian term, Therefore, the posterior $q(\mathcal{F}^{-1}(\mathbf{x}_k^f)|\mathbf{x}_{k-1}^t)$ remains gaussian. The posterior mean is $\sqrt{\alpha_k^f}\mathbf{x}_{k-1}^t$ and the standard deviation is $(\sqrt{\beta_k^f}\sqrt{1-\lambda}\mathcal{F}^{-1}\mathbf{G})(\sqrt{\beta_k^f}\sqrt{1-\lambda}\mathcal{F}^{-1}\mathbf{G})^T = \beta_k^f(1-\lambda)\mathcal{F}^{-1}\mathbf{G}\mathbf{G}^T(\mathcal{F}^{-1})^T$. Here $\mathcal{F}^{-1}$ and $(\mathcal{F}^{-1})^T$ denotes a matrix $\mathbf{U}$ due to the linearity of transformation $\mathcal{F}$, *i.e.*, $\mathcal{F}(\mathbf{x}) = \mathbf{U}\mathbf{x}$.

**Posterior of Eq.8**: Eq.8 presents $\mathbf{x}_k^t = \sqrt{\alpha_k^t}\mathcal{F}^{-1}(\mathbf{x}_k^f) + \sqrt{\beta_k^t}\sqrt{\lambda}\boldsymbol{\epsilon}_k^t$, $\boldsymbol{\epsilon}_k^t \sim \mathcal{N}(0, \mathbf{I})$. $\mathbf{x}_k^t$ is a linear transformation of $\mathcal{F}^{-1}(\mathbf{x}_k^f)$ with a Gaussian term $\sqrt{\beta_k^t}\sqrt{\lambda}\boldsymbol{\epsilon}_k^t$. Therefore, the posterior is still Gaussian with mean $\sqrt{\alpha_k^t}\mathcal{F}^{-1}(\mathbf{x}_k^f)$ and standard deviation $\beta_k^t\lambda\mathbf{I}$.

**Proof of Eq.10** (the relationship between $\mathbf{x}_k^t$ and $\mathbf{x}_k^0$ in the forward process):

From Eq.9:

$$\mathbf{x}_k^t = \sqrt{\alpha_k^t\alpha_k^f}\mathbf{x}_{k-1}^t + \sqrt{\alpha_k^t(1-\alpha_k^f)}\sqrt{1-\lambda}\mathcal{F}^{-1}(\mathbf{G}\boldsymbol{\epsilon}_k^f) + \sqrt{1-\alpha_k^t}\sqrt{\lambda}\boldsymbol{\epsilon}_k^t \tag{21}$$

we can get:

$$\mathbf{x}_k^t = \sqrt{\bar{\alpha}_k^t\bar{\alpha}_k^f}\mathbf{x}_0^t + \sqrt{1-\lambda}\sum_{s=1}^{k}\sqrt{\beta_s^f}\sqrt{\frac{\bar{\alpha}_k^t}{\bar{\alpha}_{s-1}^t}}\sqrt{\frac{\bar{\alpha}_k^f}{\bar{\alpha}_s^f}}\mathcal{F}^{-1}(\mathbf{G}\boldsymbol{\epsilon}_s^f) + \sqrt{\lambda}\sum_{s=1}^{k}\sqrt{\beta_s^t}\sqrt{\frac{\bar{\alpha}_k^t}{\bar{\alpha}_s^t}}\boldsymbol{\epsilon}_s^t \tag{22}$$

*Proof.* We proceed by mathematical induction.

For $k = 1$:

$$\begin{aligned}\mathbf{x}_1^t &= \sqrt{\alpha_1^t\alpha_1^f}\mathbf{x}_0^t + \sqrt{\alpha_1^t(1-\alpha_1^f)}\sqrt{1-\lambda}\mathcal{F}^{-1}(\mathbf{G}\boldsymbol{\epsilon}_1^f) + \sqrt{1-\alpha_1^t}\sqrt{\lambda}\boldsymbol{\epsilon}_1^t \\ &= \sqrt{\bar{\alpha}_1^t\bar{\alpha}_1^f}\mathbf{x}_0^t + \sqrt{1-\lambda}\sqrt{\beta_1^f}\sqrt{\alpha_1^f}\mathcal{F}^{-1}(\mathbf{G}\boldsymbol{\epsilon}_1^f) + \sqrt{\lambda}\sqrt{\beta_1^t}\boldsymbol{\epsilon}_1^t \\ &= \sqrt{\bar{\alpha}_1^t\bar{\alpha}_1^f}\mathbf{x}_0^t + \sqrt{1-\lambda}\sum_{s=1}^{1}\sqrt{\beta_1^f}\sqrt{\frac{\bar{\alpha}_1^t}{\bar{\alpha}_0^t}}\sqrt{\frac{\bar{\alpha}_1^f}{\bar{\alpha}_1^f}}\mathcal{F}^{-1}(\mathbf{G}\boldsymbol{\epsilon}_1^t) + \sqrt{\lambda}\sum_{s=1}^{1}\sqrt{\beta_1^t}\sqrt{\frac{\bar{\alpha}_1^t}{\bar{\alpha}_1^t}}\boldsymbol{\epsilon}_1^t\end{aligned} \tag{23}$$

Therefore, Eq.22 holds when $k = 1$. Suppose Eq.22 holds when $k = m$, *i.e.*,

$$\mathbf{x}_m^t = \sqrt{\bar{\alpha}_m^t\bar{\alpha}_m^f}\mathbf{x}_0^t + \sqrt{1-\lambda}\sum_{s=1}^{m}\sqrt{\beta_s^f}\sqrt{\frac{\bar{\alpha}_m^t}{\bar{\alpha}_{s-1}^t}}\sqrt{\frac{\bar{\alpha}_m^f}{\bar{\alpha}_s^f}}\mathcal{F}^{-1}(\mathbf{G}\boldsymbol{\epsilon}_s^f) + \sqrt{\lambda}\sum_{s=1}^{m}\sqrt{\beta_s^t}\sqrt{\frac{\bar{\alpha}_m^t}{\bar{\alpha}_s^t}}\boldsymbol{\epsilon}_s^t \tag{24}$$

For $k = m + 1$:

$$\begin{aligned}\mathbf{x}_{m+1}^t &= \sqrt{\alpha_{m+1}^t\alpha_{m+1}^f}\mathbf{x}_m^t + \sqrt{\alpha_{m+1}^t(1-\alpha_{m+1}^f)}\sqrt{1-\lambda}\mathcal{F}^{-1}(\mathbf{G}\boldsymbol{\epsilon}_{m+1}^f) + \sqrt{1-\alpha_{m+1}^t}\sqrt{\lambda}\boldsymbol{\epsilon}_{m+1}^t \\ &= \sqrt{\alpha_{m+1}^t\alpha_{m+1}^f}(\sqrt{\bar{\alpha}_m^t\bar{\alpha}_m^f}\mathbf{x}_0^t + \sqrt{1-\lambda}\sum_{s=1}^{m}\sqrt{\beta_s^f}\sqrt{\frac{\bar{\alpha}_m^t}{\bar{\alpha}_{s-1}^t}}\sqrt{\frac{\bar{\alpha}_m^f}{\bar{\alpha}_s^f}}\mathcal{F}^{-1}(\mathbf{G}\boldsymbol{\epsilon}_s^f)\sqrt{\lambda}\sum_{s=1}^{m}\sqrt{\beta_s^t}\sqrt{\frac{\bar{\alpha}_m^t}{\bar{\alpha}_s^t}}\boldsymbol{\epsilon}_s^t) \\ &\quad + \sqrt{\alpha_{m+1}^t(1-\alpha_{m+1}^f)}\sqrt{1-\lambda}\mathcal{F}^{-1}(\mathbf{G}\boldsymbol{\epsilon}_{m+1}^f) + \sqrt{1-\alpha_{m+1}^t}\sqrt{\lambda}\boldsymbol{\epsilon}_{m+1}^t \\ &= \sqrt{\bar{\alpha}_{m+1}^t\bar{\alpha}_{m+1}^f}\mathbf{x}_0^t + \sqrt{1-\lambda}\sum_{s=1}^{m+1}\sqrt{\beta_s^f}\sqrt{\frac{\bar{\alpha}_{m+1}^t}{\bar{\alpha}_{s-1}^t}}\sqrt{\frac{\bar{\alpha}_{m+1}^f}{\bar{\alpha}_s^f}}\mathcal{F}^{-1}(\mathbf{G}\boldsymbol{\epsilon}_s^f) + \sqrt{\lambda}\sum_{s=1}^{m+1}\sqrt{\beta_s^t}\sqrt{\frac{\bar{\alpha}_{m+1}^t}{\bar{\alpha}_s^t}}\boldsymbol{\epsilon}_s^t\end{aligned} \tag{25}$$

Therefore, Eq.22 holds for arbitrary $k$.  □

**Posterior of Eq.10**: Eq.10 indicates $\mathbf{x}_k^t$ is linear transformation of $\mathbf{x}_0^t$ with two groups of independent gaussian noise. Therefore, $q(\mathbf{x}_k^t|\mathbf{x}_0^t)$ is still gaussian with the standard deviation of the sum of two the two groups of gaussian noise.

## A.2 DETAILS IN THE REVERSE PROCESS

**Noise prior in the reverse process.** At the end of the forward process, $\bar{\alpha}_k^t, \bar{\alpha}_k^f \to 0$, therefore, the mean of $\mathbf{x}_k^t$ is 0. For the standard deviation term, it is the linear combination of two independent gaussian noises, so the reverse process starts from $\mathbf{x}_T = \sqrt{\lambda}\boldsymbol{\epsilon}^t + \sqrt{1-\lambda}\mathcal{F}^{-1}(\mathbf{G}\boldsymbol{\epsilon}^f), \boldsymbol{\epsilon}^t, \boldsymbol{\epsilon}^f \sim \mathcal{N}(0, \mathbf{I})$.

**Loss Function Details** In this part, we give a proof that minimizing KL-divergence between the estimated posterior and true posterior in the time-frequency diffusion model is equivalent to minimizing the sum of time noise error and frequency noise error.

*Proof.* A single time and frequency forward diffusion step can be written as

$$\mathbf{x}_k = \sqrt{\alpha_k^t \alpha_k^f}\, \mathbf{x}_{k-1} + \underbrace{\sqrt{\alpha_k^f(1-\alpha_k^t)}\, \boldsymbol{\epsilon}_k^t + \sqrt{\beta_k^f}\, \mathbf{T}\, \boldsymbol{\epsilon}_k^f}_{\boldsymbol{\eta}_k}, \qquad \boldsymbol{\epsilon}_k^t, \boldsymbol{\epsilon}_k^f \sim \mathcal{N}(\mathbf{0}, \mathbf{I}), \tag{26}$$

where $\mathbf{T} := \mathcal{F}^{-1}\mathbf{G}$. From Eq.26, it is clear that $\boldsymbol{\eta}_k \sim \mathcal{N}(\mathbf{0}, \sigma_k^2\mathbf{I})$, where $\sigma_k^2 = \alpha_k^f(1-\alpha_k^t) + \beta_k^f$.

As in DDPM (Ho et al., 2020), both the true and model reverse kernels are Gaussian with shared covariance:

$$q(\mathbf{x}_{k-1} \mid \mathbf{x}_k, \mathbf{x}_0) = \mathcal{N}(\boldsymbol{\mu}_q, \tilde{\beta}_k\mathbf{I}), \qquad p_\theta(\mathbf{x}_{k-1} \mid \mathbf{x}_k) = \mathcal{N}(\boldsymbol{\mu}_\theta, \tilde{\beta}_k\mathbf{I}), \tag{27}$$

Since the Gaussian posteriors in Eq.27 share the same covariance, the per-step KL divergence reduces to a quadratic form in the mean difference:

$$\mathrm{KL}\big(q \parallel p_\theta\big) = \frac{1}{2}(\boldsymbol{\mu}_\theta - \boldsymbol{\mu}_q)^\top (\tilde{\beta}_k\mathbf{I})^{-1}(\boldsymbol{\mu}_\theta - \boldsymbol{\mu}_q) \propto \|\boldsymbol{\eta}_k - \hat{\boldsymbol{\eta}}_k^{(\lambda)}\|_2^2. \tag{28}$$

Expanding $\|\boldsymbol{\eta}_k - \hat{\boldsymbol{\eta}}_k^{(\lambda)}\|_2^2$ with $\mathbf{T}$ isometry and independence of $\boldsymbol{\epsilon}_k^t, \boldsymbol{\epsilon}_k^f$ yields

$$\mathbb{E}\big[\|\boldsymbol{\eta}_k - \hat{\boldsymbol{\eta}}_k^{(\lambda)}\|_2^2\big] = \alpha_k^f(1-\alpha_k^t)\,\mathbb{E}\big[\|\boldsymbol{\epsilon}_k^t - \hat{\boldsymbol{\epsilon}}_k^t\|_2^2\big] + \beta_k^f\,\mathbb{E}\big[\|(\boldsymbol{\epsilon}_k^f - \hat{\boldsymbol{\epsilon}}_k^f)\|_2^2\big] \tag{29}$$

This means minimizing KL-divergence between the estimated posterior and true posterior in the time-frequency diffusion model is equivalent to minimizing the sum of time noise error and frequency noise error. And we additionally add a consistency loss as the regularization term. □

## A.3 SCALING MATRIX FOR DIFFERENT TRANSFORMATIONS

For different kinds of transformations, as described in Sec. 2.2, the scaling matrix serves two purposes:

1. Energy preservation, *i.e.,* ensuring Parseval's theorem holds: $\|\mathbf{x}\|_2^2 = \|\mathcal{F}(\mathbf{x})\|_2^2$
2. Isotropy of the injected noise: $\boldsymbol{\epsilon} \sim \mathcal{N}(0, \mathbf{I}) \to \mathrm{Cov}(\mathcal{F}^{-1}(\mathbf{G}\boldsymbol{\epsilon})) = \mathbf{I}$

In our implementation, the energy balance between the time and frequency domains has already been guaranteed. Therefore, the key point is to ensure the isotropy of the noise. As we use linear transformations, $\boldsymbol{\epsilon} \sim \mathcal{N}(0, \mathbf{I}) \to \mathrm{Cov}(\mathcal{F}^{-1}(\mathbf{G}\boldsymbol{\epsilon})) = \mathbf{I}$ means $(\mathcal{F}^{-1}\mathbf{G})(\mathcal{F}^{-1}\mathbf{G})^T = \mathbf{I}$, *i.e.,* $\mathbf{G}\mathbf{G}^T = \mathbf{F}\mathbf{F}^T$.

For DFT, we use the same implementation as (Crabbé et al., 2024), therefore, we use the same scaling matrix in 2.2. For DCT and DWT, our implementations ensure they are unitary transformations,*i.e.,* their corresponding transform matrices are orthogonal (in the real domain). Therefore, $\mathbf{F}\mathbf{F}^T = \mathbf{I}$, so we set $\mathbf{G} = \mathbf{I}$ for simplicity.

## A.4 ARCHITECTURE DETAILS

Fig.3 presents the diffusion backbone of our denoising block. The block adopts a time-frequency coupled dual-branch residual The input sequence is first projected using a $1 \times 1$ convolution followed by a ReLU activation. Time and Frequency diffusion step embeddings are generated via fully connected layers with SiLU activations and then concatenated with external side information.

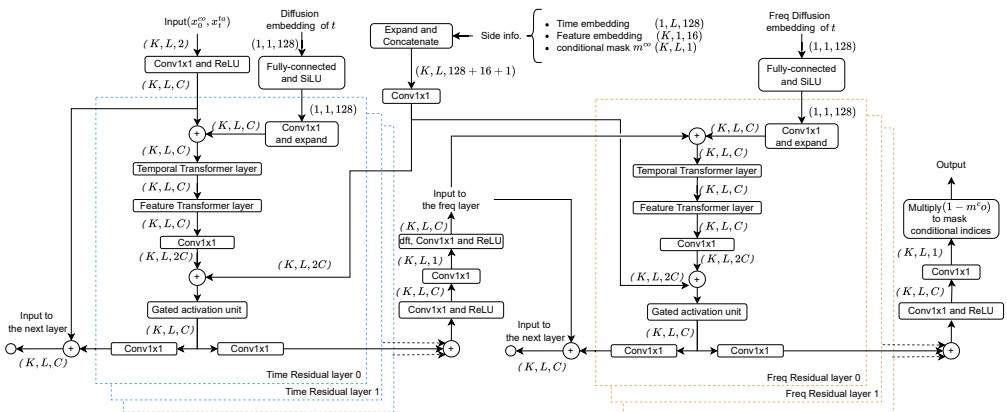

Figure 3: Architecture details of our denoising model $\epsilon_\theta$.

During residual modeling, the network first processes the representation in the time-domain branch. After an initial $1 \times 1$ convolution and ReLU projection, the features are passed through a temporal Transformer layer and a feature Transformer layer to capture inter-channel and intra-channel dependencies. The outputs are fused using a gated activation unit (GAU) and further refined by stacked $1 \times 1$ convolutions. Residual connections feed the results back into the input stream to form the input for the next layer.

Then the time-domain representation is then fed into the frequency-domain branch with a similar structure to get the frequency domain output,. By processing in a time-to-frequency order, the architecture establishes a complementary relationship between global trend modeling and fine-grained frequency refinement.

## A.5 DATASET DETAILS

The Physionet2012 dataset (Silva et al., 2012) consists of 4000 irregularly-sampled medical time series data including 35 variables (*e.g.,* Albumin and heart rate) collected from ICU with a total length of 48 hours. Consistent with previous studies (Tashiro et al., 2021), the dataset is processed hourly to get 48 timesteps. The processed dataset contains nearly 80% originally missing values without ground truth. In our experiments, we random select 10/50/90% of the observed values as the imputation targets (*i.e.,* ground truth of test dataset).

The air quality dataset contains PM2.5 data collected from 36 monitor stations in Beijing. All the air quality data are collected hourly for 12 months. The original dataset contains 13.3% missing values with artificial ground-truth with a non-random missing pattern.

The details of the datasets are presented in Tab.4.

Table 4: Details of Physionet2012 and Air Quality datasets.

| Dataset | # Samples | # Features | Time Steps | Missing Type | Missing Ratio |
|---|---|---|---|---|---|
| PhysioNet | 4000 | 35 | 48 | Originally Missing & Random | 80% (Original) |
| Air Quality | 5633 | 36 | 36 | Non-Random & Artificial | 13% (Original) |

## A.6 EVALUATION METRICS

In this section, we present the details of evaluation metrics in our experiments. $y, \hat{y} \in \mathbb{R}^{K \times L}$ denote the ground truth and output of our model and $M$ is the indicator matrix.

**Mean Absolute Error (MAE)** calculates the average $L_1$ error between the imputed samples and the ground truth of the time series:

$$\mathbf{MAE}(y, \hat{y}) = \frac{\sum_{i=1}^{K} \sum_{j=1}^{L} M_{ij} |y_{ij} - \hat{y}_{ij}|_1}{\sum_{i=1}^{K} \sum_{j=1}^{L} M_{ij}} \tag{30}$$

MAE reflects the overall deviation across all points, emphasizes the overall accuracy of the model outputs. Compared with RMSE, MAE is more robust to outliers in the data.

**Rooted Mean Square Error (RMSE)** calculates the average $L_2$ error between the imputed samples and the ground truth of time series:

$$\text{RMSE}(y, \hat{y}) = \sqrt{\frac{\sum_{i=1}^{K} \sum_{j=1}^{L} M_{ij} \|y_{ij} - \hat{y}_{ij}\|_2^2}{\sum_{i=1}^{K} \sum_{j=1}^{L} M_{ij}}} \tag{31}$$

RMSE highlights potential large errors, so a few big deviations may dominate the score. Compared with MAE, RMSE is less robust to outliers but better captures worst-case performance.

## A.7 EXPERIMENT DETAILS

### A.7.1 HYPERPARAMETERS

The hyperparameters in our experiments is detailed in Table.5. And the choice of $\lambda$ according to datasets and transformations is presented in Table.6

Table 5: Hyperparameter details in our experiments

| Dataset | Physionet | Air Quality |
|---|---|---|
| Epochs | 200 | 200 |
| Batch Size | 16 | 16 |
| Learning Rate | 0.001 | 0.001 |
| #Time Layers | 4 | 4 |
| #Freq Layers | 4 | 4 |
| Channels | 128 | 128 |
| #Time Heads | 8 | 8 |
| #Freq Heads | 8 | 8 |
| #Time Diffusion Embedding Dim | 128 | 128 |
| #Freq Diffusion Embedding Dim | 128 | 128 |
| $\beta_0^t$ | 0.0001 | 0.0001 |
| $\beta_1^t$ | 0.5 | 0.5 |
| $\beta_0^f$ | 0.0001 | 0.0001 |
| $\beta_1^f$ | 0.25 | 0.25 |
| $\omega$ | 0.2 | 0.2 |
| #Steps | 50 | 50 |
| Noise Scheduler | quad | quad |
| #Time Embedding Dim | 128 | 128 |
| #Feature Embedding Dim | 16 | 16 |

Table 6: Choices of $\lambda$ on different datasets and transformations,

| | Physionet 10% | Physionet 50% | Physionet 90% | Air Quality |
|---|---|---|---|---|
| DFT | 0.25 | 0.25 | 0.75 | 0.75 |
| DWT | 0.25 | 0.75 | 0.75 | 0.75 |
| DCT | 0.25 | 0.75 | 0.75 | 0.75 |

### A.7.2 VISUALIZATION RESULTS

The imputation results on Air Quality dataset and Physionet2012 dataset with missing ratio 10%, 50% and 90% are presented in Fig.4,5,6,7.

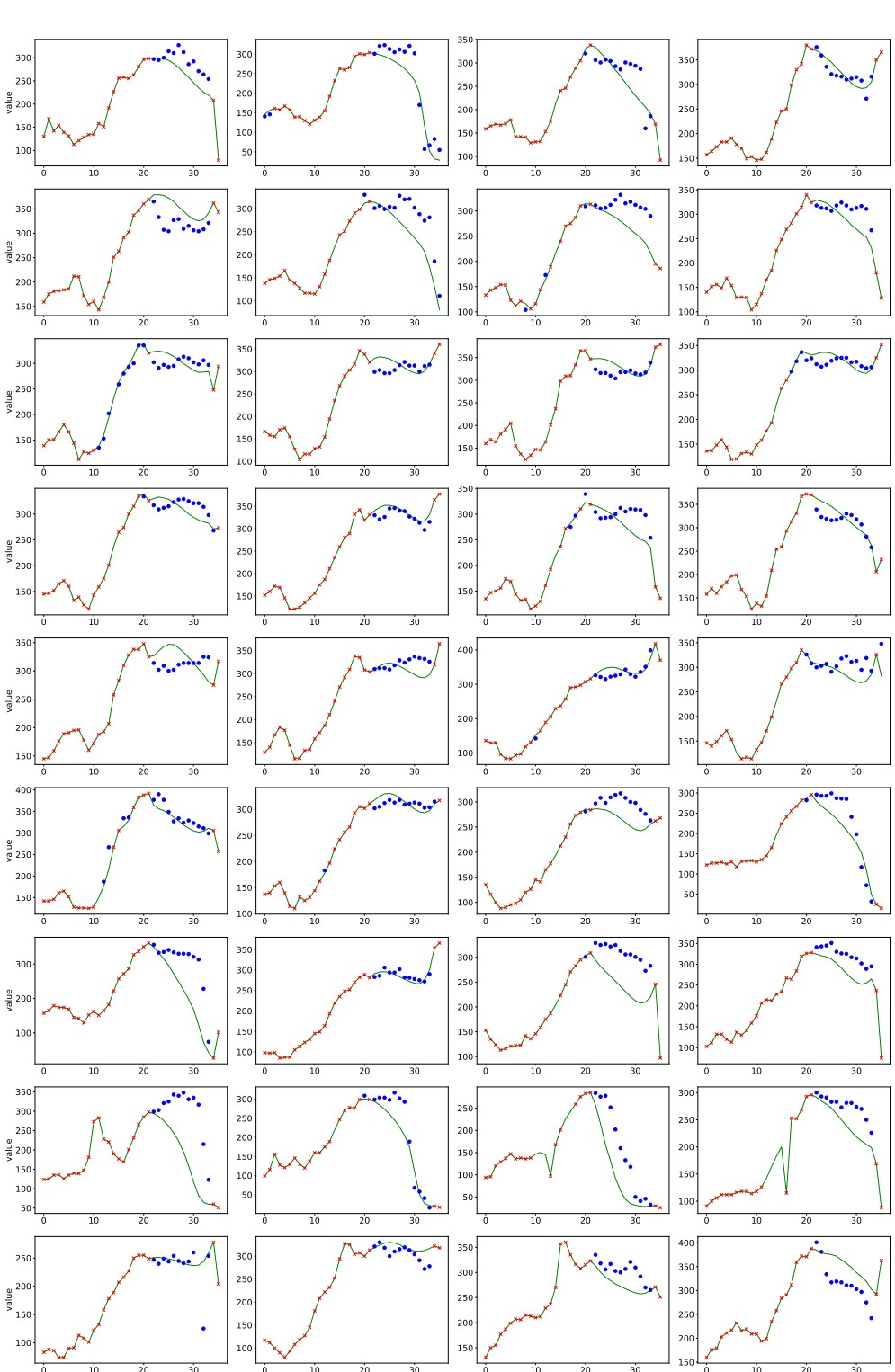

Figure 4: Visualization of imputation results on AQI dataset from Channel 1 to Channel 36. The solid line represents the imputation results, the blue dots represent the ground truth of the missing points, and the red crosses represent the observed values.

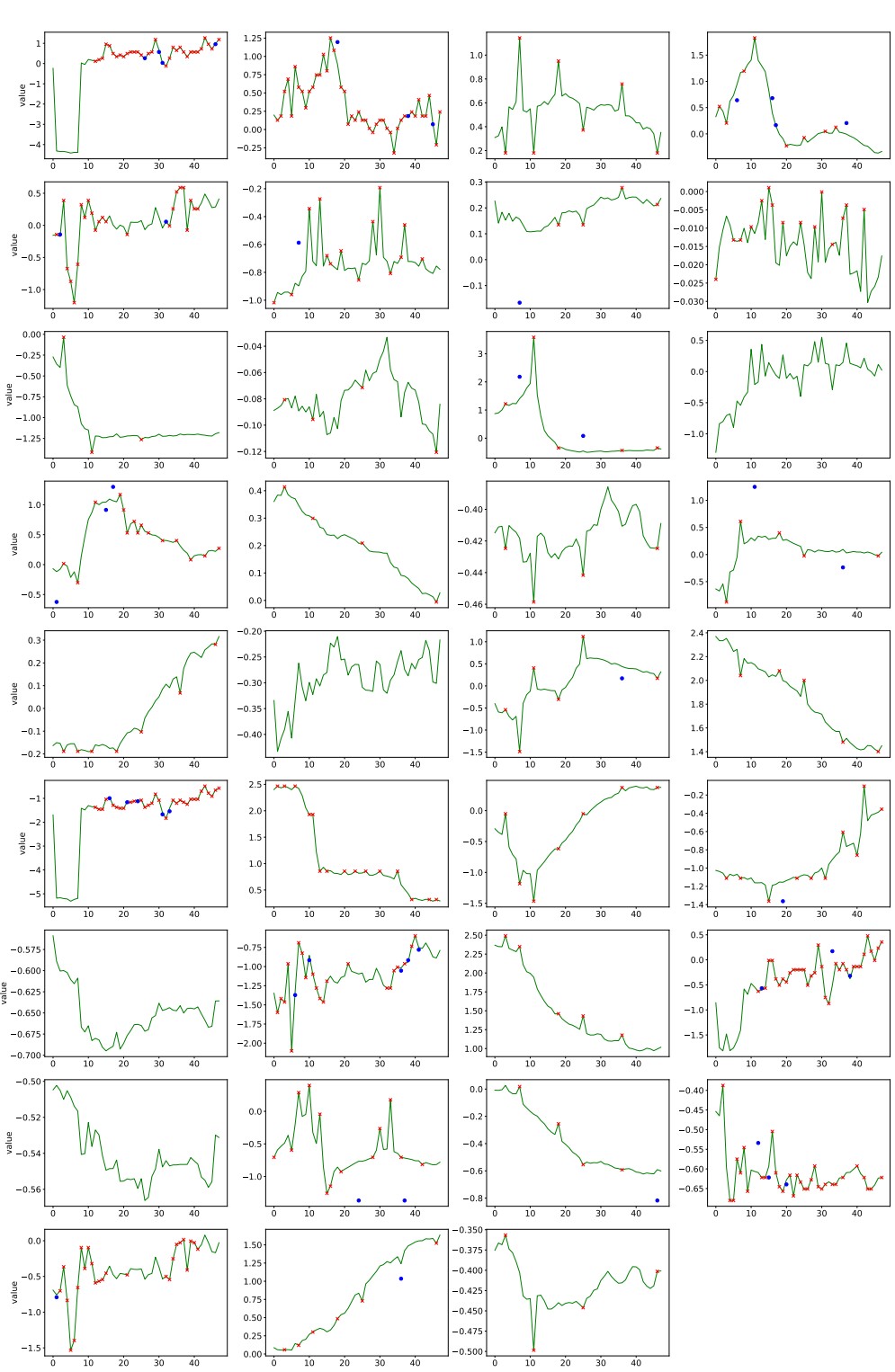

Figure 5: Visualization of imputation results on Physionet2012 dataset from Channel 1 to Channel 35. The solid line represents the imputation results, the blue dots represent the ground truth of the missing points, and the red crosses represent the observed values.

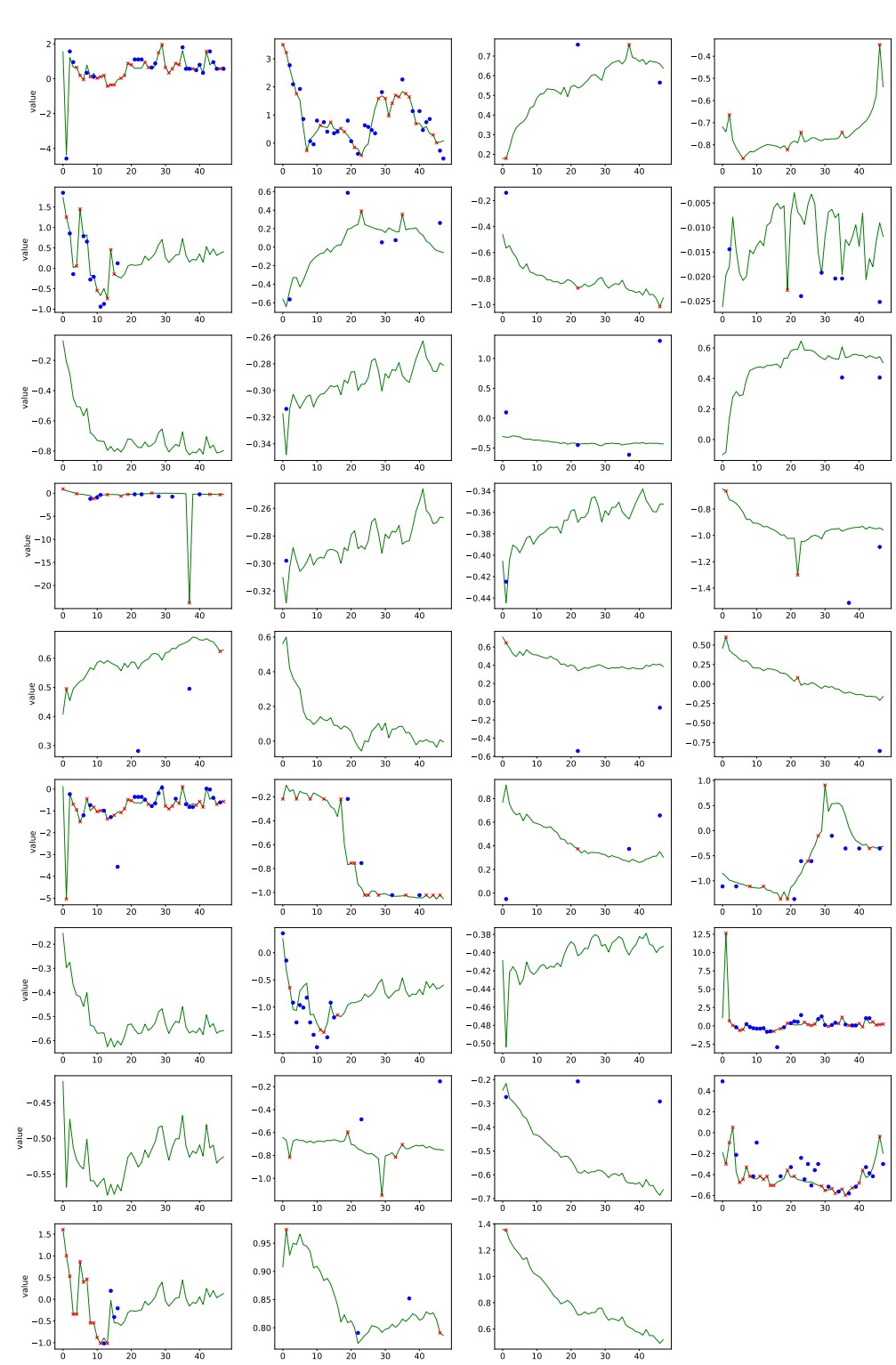

Figure 6: Visualization of imputation results on Physionet2012 dataset from Channel 1 to Channel 35 with 50% missing. The solid line represents the imputation results, the blue dots represent the ground truth of the missing points, and the red crosses represent the observed values.

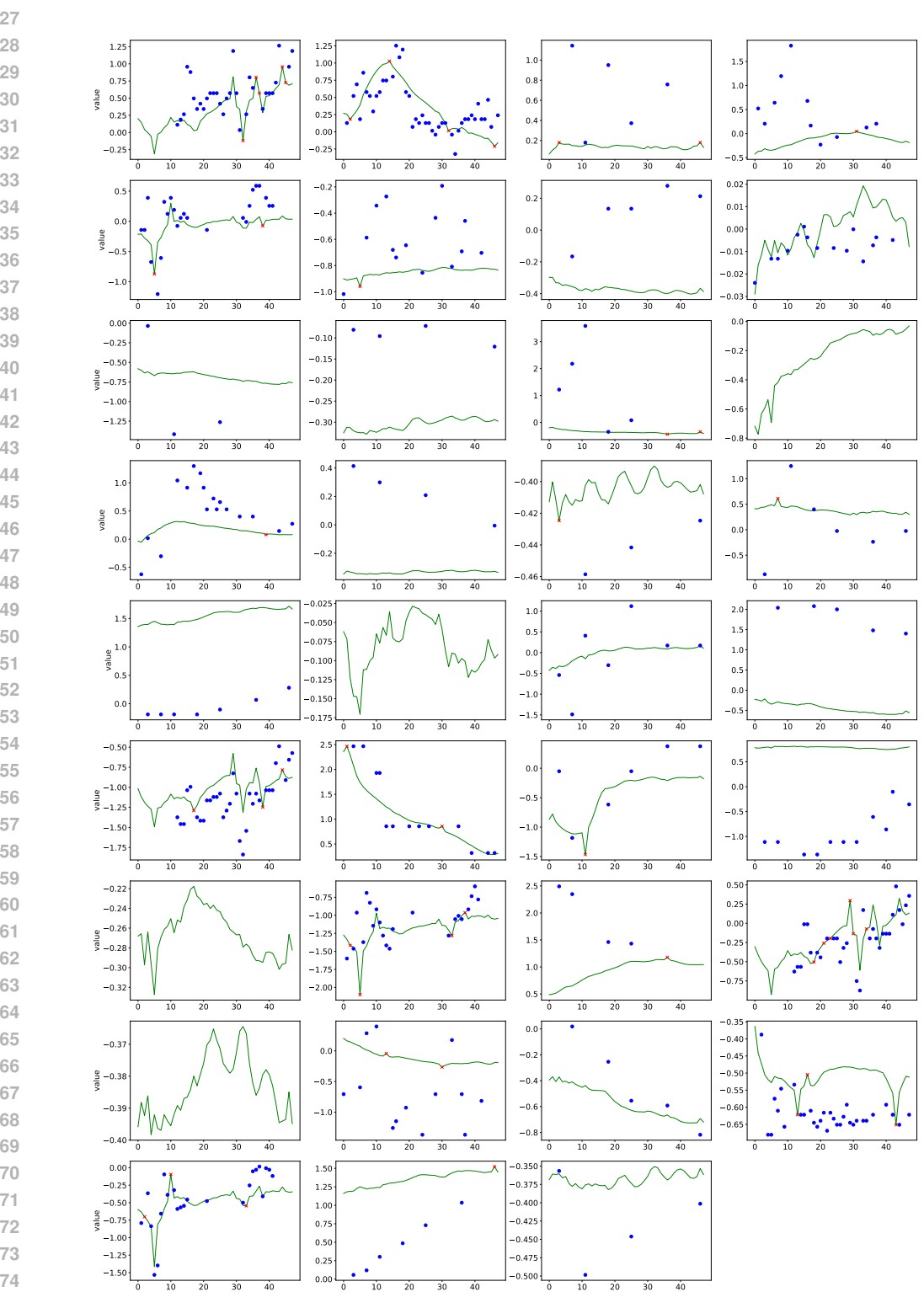

Figure 7: Visualization of imputation results on Physionet2012 dataset from Channel 1 to Channel 35 with 90% missing. The solid line represents the imputation results, the blue dots represent the ground truth of the missing points, and the red crosses represent the observed values.

