# OpenReview forum: "TFD: Spectrally Guided Time–Frequency Diffusion Model For Time Series Imputation"
_ICLR.cc/2026/Conference — Submitted to ICLR 2026_

### Official Review · Reviewer_PYSJ · 2025-10-29

**Soundness:** 2
**Presentation:** 3
**Contribution:** 2
**Rating:** 4
**Confidence:** 3

**Summary:**

This paper proposes TFD (Time-Frequency Diffusion), a hybrid diffusion model for time series imputation that addresses the inability of time-domain-only diffusion models to reconstruct high-frequency details. The key innovation is a frequency-aware high-pass timestep embedding that provides spectral guidance by emphasizing relevant frequency bands at specific denoising steps.

**Strengths:**

1. The paper is well-writen and easy to understand.

2. The paper makes a solid theoretical contribution by extending diffusion models to hybrid time-frequency domains with formal mathematical grounding and addresses important limitations in existing methods.

3. The paper provides theoretical analysis to demonstrate the effectiveness of the method.

4. It claims state-of-the-art performance across multiple datasets and scenarios for time series imputation.

**Weaknesses:**

1. The structure of this paper is not clear enough. The main body includes too many unnecessary methodological details, while there is very little presentation of experimental results, making it difficult to fully demonstrate the effectiveness of the proposed method.

2. Important behavioral questions remain unanswered: (a) Does the time-domain branch indeed focus on low-frequency components and the frequency branch on high-frequency components as claimed? This could be validated through spectral analysis of branch outputs. (b) How do errors propagate between branches given their sequential dependency? (c) How does the frequency-aware embedding actually modulate the denoising process at different steps? (d) What happens when training data violates the low-pass assumption? Understanding model behavior is crucial for both practical application and future research directions.

3. Proposition 3.1 assumes time series data follows a low-pass spectral characteristic, but this assumption is neither validated on the experimental datasets nor analyzed for different time-series types. Real-world time series can exhibit diverse spectral characteristics—financial data may have heavy tails, physiological signals may have dominant high-frequency components, and non-stationary processes may violate this assumption entirely.

**Questions:**

1. Does the low-pass assumption hold empirically for your datasets? How does performance degrade for time series with different spectral characteristics?

---

### Official Review · Reviewer_zjxw · 2025-10-30

**Soundness:** 2
**Presentation:** 2
**Contribution:** 2
**Rating:** 2
**Confidence:** 4

**Summary:**

This paper proposes TFD (Time-Frequency Diffusion), a hybrid diffusion model designed for time-series imputation. The framework employs a unique coarse-to-fine reconstruction strategy: noise is injected first in the frequency domain and then in the time domain, while the denoising process proceeds in the reverse order (time to frequency). TFD utilizes a dual-branch (time/frequency) architecture based on CSDI and introduces a frequency-aware high-pass timestep embedding aimed at emphasizing high-frequency components during the final restoration stages. The model reports performance using MAE/RMSE metrics on the PhysioNet 2012 and Air Quality datasets.

**Strengths:**

- The rationale for the hybrid time-frequency approach is clear and well-motivated. Both the forward and backward diffusion processes are explicitly defined with accompanying mathematical formulas and algorithms.

- Key architectural decisions, including the spectral backends (e.g., DFT, DCT, Haar‑DWT) and the use of a consistency loss, are clearly specified.

**Weaknesses:**

- The empirical validation is insufficient, relying on only two datasets and two non-spectral metrics (MAE/RMSE). This narrow focus provides an inadequate assessment of the model's performance across diverse data types, missingness patterns, and real-world complexities.

- The reported performance improvements over existing models are modest. Critically, the absence of multi-seed results or error bars prevents any assessment of statistical significance, making the claimed superiority unreliable.

- The core motivation relies on an unverified spectral assumption whose validity is not tested.

- The exact implementation formula for the high-pass timestep embedding is missing key information. Specifically, it is unclear whether the constants $\alpha$, $\gamma$, and $\sigma$ from Eq. 20 are treated as fixed hyperparameters, learned parameters, or simply fixed constants, which critically impacts the replicability and analysis of the method.

- The paper strongly claims the ability to restore high-frequency details, yet the evaluation is based solely on MAE and RMSE metrics, which are known to primarily reflect low-frequency fidelity and average error. The paper lacks the necessary supporting spectral analysis (e.g., using spectral density or tail-sensitive metrics) to substantiate its main claim.

- The paper fails to clearly explain how noise is balanced between the two diffusion branches. Furthermore, it lacks experimental justification for crucial design choices, such as the decision to separate the branches during training, and provides no analysis of computational speed or efficiency, which is vital for dual-branch models.

**Questions:**

Please refer to the Weaknesses part above.

---

### Official Review · Reviewer_sPqa · 2025-10-30

**Soundness:** 3
**Presentation:** 3
**Contribution:** 3
**Rating:** 6
**Confidence:** 3

**Summary:**

This paper proposes a time series interpolation method based on a diffusion model, focusing on the destruction and reconstruction of time series information from two levels: time domain and frequency domain.

**Strengths:**

1. Intuitive insight: Combining the characteristics of diffusion models and time series.

2. Theoretical analysis: Proposition 3.1 provides valuable insights while making it easy to understand the author's motivations.

**Weaknesses:**

1. Excessive hyperparameter settings are involved in this method, which involves hyperparameters such as $\lambda$ and the selection of frequency transformation. On the one hand, there is no adaptive method to adjust them. On the other hand, Table 3 shows that in some cases, the selection of lambda does not have a significant impact. The author's method does not improve significantly compared to the method that degenerates to only consider time-domain or frequency-domain information.


2. Some highly relevant methods that modified the adding noise process of time series diffusion have not been considered, such as [1]

[1] Wang C, Yang L, Wang Z, et al. A Non-isotropic Time Series Diffusion Model with Moving Average Transitions[C]//Forty-second International Conference on Machine Learning.

**Questions:**

1. The author has repeatedly emphasized the lack of integration between time-domain and frequency-domain information in current methods, but mainly emphasizes the importance of frequency-domain information. I would like to know how the author's method reflects the combination of the two, or in other words, what problems would exist if I only consider frequency-domain information for diffusion? Why the time domain information can not be replaced by low-frequency information?

2. Does the order of adding noise have to be frequency domain first and then time domain? What changes will occur if it is reversed?

3. The assumption $\left|\hat{\mathbf{x}}_t(\omega)\right|^2 \propto \omega^{-\alpha}$ in Proposition 3.1 seems reasonable, but is it possible that there may be situations where this assumption is not met, such as when the high-frequency energy of economic data is generally high? I am concerned about the practicality of the paper's method in this type of data, and whether there may be an adaptive tuning mechanism.

---

### Official Review · Reviewer_76QT · 2025-11-03

**Soundness:** 1
**Presentation:** 2
**Contribution:** 1
**Rating:** 2
**Confidence:** 4

**Summary:**

The paper proposes TFD, a hybrid time–frequency diffusion model for time-series imputation. The method extends a diffusion-based model for imputation such as CSDI by introducing a frequency-domain branch and a frequency-aware timestep embedding.

**Strengths:**

* The paper is written clearly.

* The high-level idea of incorporating frequency guidance into diffusion models is intuitively interesting.

**Weaknesses:**

* The experimental section lacks scientific rigor. The paper does not rerun or verify prior baselines, making performance comparisons invalid. Most baseline results are directly copied from previous papers: all results are copied from [1] except the CSBI baseline which is copied from [2]. This is not clarified in the paper.

* The original CSBI paper only shows results for Air Quality, Physionet 10% and Physionet 50% with 2 decimal places. In Table 2, the same values appear but with additional decimals. Moreover, Physionet with 90% missingness was not reported in the original paper. Where was this value obtained from?

* The proposed method relies on DFT, DCT, and DWT to perform diffusion in the frequency domain. However, these transforms require fully observed, uniformly sampled sequences. Since the imputation task explicitly assumes missing data, applying these transforms directly is mathematically inconsistent. This undermines the technical validity of the proposed model and its claimed advantages over prior methods designed for missing data (e.g., LSCD which explicitly used Lomb–Scargle to estimate power spectrum, which does not rely on uniformly sampled or complete time series.).

* Claims of SOTA performance are overstated. As noted in the ablation studies, no single choice of frequency-domain transformation consistently improves results across datasets.

* There does not seem to be any information/comparison about runtime performance.

[1] Fons, E., Sztrajman, A., El-Laham, Y., Ferrer, L., Vyetrenko, S. &amp; Veloso, M.. (2025). LSCD: Lomb–Scargle Conditioned Diffusion for Time series Imputation. ICML 2025.
[2] Yu Chen, Wei Deng, Shikai Fang, Fengpei Li, Nicole Tianjiao Yang, Yikai Zhang, Kashif Rasul, Shandian Zhe, Anderson Schneider, and Yuriy Nevmyvaka. Provably convergent Schrödinger bridge with applications to probabilistic time series imputation. ICML 2023.

**Questions:**

* Did you rerun all baselines under the same setup, or did you copy numbers from prior publications? If copied, how did you ensure comparability?

* How were CSBI results derived for datasets where the original paper reported none?

* How did you handle missing data prior to applying frequency-domain transformations (DFT/DCT/DWT)?

---

### Meta-Review · Area_Chair_YVLr · 2026-01-06

**Summary:**

Four reviewers gave detailed comments on the paper, and their major concerns include: 1) The writing and the organization of the paper should be improved. 2) Some assumptions are not validated, and motivation should be further clarified. 3) The evaluation should be further improved by adding more explanations, comparing with more recent baselines, and less significant performance improvement.
Based on reviewers' comments, I suggest to reject the paper.

**Reviewer Concerns:**

There is no rebuttal as the authors did not response to the comments.

**Reviewer Scores:**

There is no discussion.

---

### Decision · Program_Chairs · 2026-01-26

Reject